# Impact of Tralopyril and Triazolyl Glycosylated Chalcone in Human Retinal Cells’ Lipidome

**DOI:** 10.3390/molecules27165247

**Published:** 2022-08-17

**Authors:** Cátia Vilas-Boas, Logan Running, Daniela Pereira, Honorina Cidade, Marta Correia-da-Silva, Gunes Ekin Atilla-Gokcumen, Diana S. Aga

**Affiliations:** 1Laboratory of Organic and Pharmaceutical Chemistry, Department of Chemical Sciences, Faculty of Pharmacy, University of Porto, 4050-313 Porto, Portugal; 2CIIMAR/CIMAR—Interdisciplinary Center for Marine and Environmental Research, University of Porto, 4450-208 Matosinhos, Portugal; 3Chemistry Department, University at Buffalo, The State University of New York, Buffalo, NY 14260, USA

**Keywords:** antifouling biocides, chalcone, cytotoxicity, Econea, human cells, LC-Q-TOF, lipidomics

## Abstract

Antifouling (AF) coatings containing booster biocides are used worldwide as one of the most cost-effective ways to prevent the attachment of marine organisms to submerged structures. Nevertheless, many of the commercial biocides, such as Econea^®^ (tralopyril), are toxic in marine environments. For that reason, it is of extreme importance that new efficient AF compounds that do not cause any harm to non-target organisms and humans are designed. In this study, we measured the half-maximal inhibitory concentration (IC_50_) of a promising nature-inspired AF compound, a triazolyl glycosylated chalcone (compound **1**), in an immortalized human retinal pigment epithelial cell line (hTERT-RPE-1) and compared the results with the commercial biocide Econea^®^. We also investigated the effects of these biocides on the cellular lipidome following an acute (24 h) exposure using liquid chromatography quadrupole time-of-flight mass spectrometry (LC-Q-TOF/MS). Our results showed that compound **1** did not affect viability in hTERT-RPE-1 cells at low concentrations (1 μM), in contrast to Econea^®^, which caused a 40% reduction in cell viability. In total, 71 lipids were found to be regulated upon exposure to 10 µM of both compounds. Interestingly, both compounds induced changes in lipids involved in cell death, membrane modeling, lipid storage, and oxidative stress, but often in opposing directions. In general, Econea^®^ exposure was associated with an increase in lipid concentrations, while compound **1** exposure resulted in lipid depletion. Our study showed that exposure to human cells at sublethal Econea^®^ concentrations results in the modulation of several lipids that are linked to cell death and survival.

## 1. Introduction

Marine biofouling corresponds to a successive accumulation of a wide range of colonizer species (e.g., bacteria, diatoms, macroalgae, tunicates, barnacles, mussels, and tubeworms) into surfaces immersed in seawater [1]. To prevent or inhibit the settlement and growth of marine organisms on underwater surfaces, AF biocides are coated on the surface and must be continually released at the surface–water interface at a rate necessary to generate a toxic concentration within the surface boundary layer. Biofouling of underwater surfaces of ships and vessels increases the frictional drag and has become the primary cost associated with the increase in fuel consumption in the marine industry [2]. For decades, the marine industry has made efforts to prevent this natural fouling process.

To date, only a few commercial biocides have the necessary combination of being environmentally safe and yet effective AF agents. Mercury and arsenic and their compounds, and organotins, are examples of effective AF agents that have been deemed unacceptable due to adverse environmental or human health risks. Early formulations of organotin-based coatings have been removed from the market due to their ecotoxicity and were replaced by copper-based coatings, boosted with organic biocides, such as Diuron, Irgarol 1051^®^, Sea Nine-211^®^, Zinc pyrithione (ZnPT), and others, to increase their effectiveness [3,4]. However, most of these organic booster biocides have also been found to persist in water and sediments at levels that are harmful to the marine environment [5]. As a result, the marine industry is facing the phase-out of current persistent, bioaccumulative, and toxic biocides, limiting the available alternatives and creating a great opportunity for the development of new AF agents.

The biocide Econea^®^ (tralopyril, Janssen PMP, Figure 1A) emerged as a metal-free biocidal additive replacement for copper and was accepted as an AF agent for inclusion under the Biocidal Products Regulation (EU, 2012) in 2014 to control the fouling of barnacles, hydroids, mussels, oysters, and polychaetes [6,7]. Despite its uncertain mode of action, Econea^®^ is thought to act as an uncoupler, interfering with routine mitochondrial functions and ATP production, with a high tendency to present toxicity to non-target species [8]. Recent studies demonstrated that Econea^®^ impacts marine invertebrates, being rapidly bioaccumulated by the mussel *Mytilus galloprovincialis* and modulating a total of 46 Mytilus proteins involved in metabolism, immune system, active efflux, and oxidative stress, causing several alterations after the depuration period [8,9]. Econea^®^ also affects the metabolism of amino acids, energy, and lipids, which was associated with the regulation of the thyroid and nervous system and tail muscle tissue in zebrafish [6,10]. With the widespread use of Econea^®^, more attention should be paid to its potential to harm the marine ecosystem and humans due to their potential for exposure to biocides during boating maintenance and/or the food chain [11,12].

Inspired by secondary metabolites of microorganisms and sessile marine organisms, a variety of nature-inspired antifoulants (NIAFs) have been synthesized and developed by [13,14,15,16,17,18]. In our previous studies, a triazolyl glycosylated chalcone (compound **1**, Figure 1B) exhibited higher bioactivity (EC50 = 3.28 μM; 2.43 µg/mL) than the commercial biocide Econea^®^ (EC_50_ = 4.012 μM; 1.40 µg/mL) against the settlement of *M. galloprovincialis* larvae; a therapeutic ratio greater than 61, and inhibitory effects on the growth of biofilm-forming microalgae, *Navicula* sp. (EC_50_ = 41.76 μM; 30.94 µg/mL) were observed [14,19]. Compound **1** was also found to be non-toxic against the non-target organism *Artemia salina*, causing less than 10% mortality at 25 and 50 μM, suggesting its potential as an eco-friendly compound for the development of new AF marine coatings [17,19].

Lipids are relevant cellular components for assessing the biological effects of compounds with a high n-octanol-water partition coefficient (log Kow), such as the AF biocides because lipids play a role in bioaccumulation lipophilic compounds. In addition, AF biocides can disrupt membranes and lipid metabolism and alter the fluidity and permeability of cell membranes of target and non-target organisms [17,20]. Biocides can impact human health not only through the food chain but also by direct exposure during boating maintenance, causing skin allergies, eye irritations, and respiratory disorders [11,21]. Lipids provide several important functions to maintain cellular homeostasis including proper mitochondrial function, energy storage, membrane support, and signaling activities. The alterations in lipid metabolism and levels can lead to adverse effects, such as cellular proliferation and death [22]. For example, ceramides (CERs) are usually linked to apoptosis while other lipids, such as phosphatidylcholines (PCs) and triacylglycerols (TAGs), are involved in membrane integrity and structural support, and storage of excess fatty acids, respectively [23]. With these important functions, lipidomics has become an important field of study, offering a powerful tool to identify changes in lipid compositions induced by harmful chemicals, and can provide insights into their mechanisms of toxicity [22,24,25].

In this study, we evaluated the toxicity of Econea^®^ and compound **1** using a non-cancerous immortalized retinal pigment epithelial cell line (hTERT-RPE-1) by characterizing the changes in its lipidome upon exposure to these AF agents. This cell line was chosen because of the lack of toxicity information available on humans for AF agents. This cell line is also relevant for this study because the eyes are one of the first organs that are exposed to aquatic contaminants and can provide a good model for an important human exposure pathway. In this study, we did not synchronize cells at a particular cell cycle stage as a diverse population is a better representative of what happens at the organismal level when target and non-target organisms are exposed to these compounds [26]. Firstly, the 3-(4,5 dimethylthiazol 2-yl)-2,5-diphenyltetrazolium bromide-based MTT assay was conducted on hTERT-RPE-1 to obtain the half-maximal inhibitory concentrations (IC_50_) for Econea^®^ and compound **1** [27,28]. Then, a lipidomic approach using liquid chromatography quadrupole time-of-flight mass spectrometry (LC-Q-TOF/MS) was employed to assess the effects of acute exposure to sublethal concentrations of Econea^®^ and compound **1**. It was observed that Econea^®^ induced cell mortality at lower concentrations (1 µM) and resulted in the accumulation of several lipid families, but not in compound **1**. Results from this study provide novel insights into the toxicity of new AF agents in human cells.

## 2. Results and Discussion

### 2.1. Cytotoxicity of Econea^®^ and Compound 1 in Retinal Human Cells

The viability of the hTERT-RPE-1 cell line was examined by MTT assay based on mitochondrial activity to determine compound **1** and Econea^®^-induced cytotoxicity at different concentrations. The results are shown in Figure 2.

After 24 h, Econea^®^ showed significant cell toxicity that caused 40% cell death even at the lowest tested concentration of 1 µM; dose-dependent toxicity resulting in 80% cell death at 100 µM was observed. Previous results showed the toxicity of Econea^®^ against several marine organisms, such as *Daphnia magna* (planktonic crustacean) [14], *Danio rerio* (fish) [26], *Chlamydomonas reinhardtii* (green alga) [27], and *M. galloprovincialis* larvae (mussel) [14,28]. Our results show that Econea^®^ is also toxic to human retinal cells, with a calculated IC_50_ of 20.29 µM (7.09 µg/mL), reinforcing the urgent need to find harmless AF alternatives. On the other hand, compound **1** exhibited a ~50% decrease in cell viability only at the highest concentration tested (100 µM) in retinal cells demonstrating that compound **1** does not cause toxicity in its range of effective concentrations against the mussel *M. galloprovincialis* (EC_50_ = 3.28 μM) [18]. These results show that compound **1** is a promising non-toxic alternative that can be used as an AF agent in marine coatings.

### 2.2. Effects of Acute Exposure on Cellular Lipidome

Lipid metabolism is an important metabolic process that provides energy and supports various physiological and developmental processes. Membrane lipids are dynamic structures that play important roles in cell functions and undergo several transient physicochemical changes to sustain cellular function under stress conditions [29]. Several AF biocides proved to be toxic and induce oxidative stress in several marine organisms, triggering alterations in their lipid profile and inducing membrane permeabilization [10,20,29,30]. To investigate the effects of compound **1** and Econea^®^ on the cellular lipidome of hTERT-RPE-1 cells, we conducted comparative lipidomics in a pairwise fashion. We treated cells with compound **1** and Econea^®^, extracted lipids, and compared the levels of lipids to untreated control cells. Fold-change values were obtained by taking average raw abundance values for the 10 µM compound **1**-treated and Econea^®^-treated cells to the average raw abundance of their corresponding untreated control groups. At 10 µM, compound **1** did not induce any significant toxicity whereas Econea^®^ induced 40% cell death. After normalization based on protein content, an analysis of 357 lipids including fatty acids (FAs), glycerolipids (DAGs and TAGs), sphingolipids (CERs, dihydroceramide (DiHCERs), deoxyceramide (DeoxyCERs), deoxy-(dihydro)ceramides (DiHDeoxyCERs), dihydrosphingomyelins (DiHSMs), SMs), and phospholipids (PCs) was performed, as we described previously [31,32,33]. These species were initially considered due to the observed responses of sphingolipids [34] and TAGS [35] during oxidative stress. We also considered PCs as these compounds make up a significant component of membranes and may be a target of AF compounds [36]. Isotopically labeled standards or non-endogenous representative lipid standards were used as additional quality controls. We included a single standard for each lipid class considered in this study. Internal standards were used to monitor for shifts in retention time (RT), loss of sensitivity, and changes in exact mass (>10 ppm error) for each sample. Errors in accurate mass measurements (ppm error) and RT values from a representative standard mixture can be seen in Appendix A. Based on internal standard RT (Appendix A), we expected FAs to be the first lipid classes to elute and appear in the initial 35 min of the chromatogram. A few FAs were detected; however, they did not exhibit any significant fold changes (*p* < 0.05) (i.e., FA C16:0 and C18:0). Overall, FAs appeared unaffected by the treatments, and no further analysis for this compound class was performed. The remaining lipids were eluted in the subsequent 40 to 70 min RT range of the chromatogram. PCs, SMs, and dHSMs were detected in the protonated form [M + H]^+^. However, in our established protocol, most DAGs were detected as [M-OH]^+^ with only two DAGs detected as [M + H]^+^ [31,32,33]. All TAGs were detected as their ammonium adduct [M + NH_4_]^+^. The rest of the lipids, CERs, DiHCERs, DeoxyCERs, and DiHDeoxyCERs were identified as [M-H]^−^ [31,32,33] (Appendix A). A total of 101 features were selected and tentatively identified based on accurate mass measurements, with a threshold of <10 ppm error. Potential features were only considered if their corresponding *m*/*z* values in the blank samples were <10% of their signals in the treated samples. The raw abundances for each detection can be seen in Appendix A.

Analysis of the detected lipids showed that the most abundant lipids in hTERT-RPE-1 cells were PCs representing 47% of the lipids analyzed, followed by TAGs (12%), CERs, DeoxyCERs, and SMs (8%), and DAGs as the sixth most abundant group (7%) (Figure 3). The rest of the lipid classes represented less than 5% of the lipids analyzed in both treatments. After Econea^®^ and compound **1** exposure, this workflow resulted in 71 species that showed a significant alteration (*p*-value < 0.05) and fold change lower (depletion) and greater (accumulation) than 1, compared to control (Figure 3A). Appendix A includes fold change values and significant alterations (*p*-values < 0.05).

Lipid accumulation was the most frequent response to Econea^®^ exposure (43), with a similar behavior already observed in proteins of the target marine organism *M. galloprovincialis* [8]. After Econea^®^ exposure, an accumulation in almost 20 PCs, followed by DAGs (7) and TAGs (6) was observed. A modest accumulation (1.5–3-fold change) was observed in saturated and unsaturated sphingolipids, which includes Cer C14:0, DiHCER C18:0, C20:0, and C22:0, DiHSMs, C14:0 and C16:0, DAG C34:2, PC C28:3, C30:0, C30:1, C32:1, C32:2, C32:3, C34:2, C34:3, C36:3, 34:2-3, TAG C30:4, C54:4, C54:5, C56:6, C56:8, and C58:8 (Figure 3A). A high accumulation (9-fold change) of DiHCER C16:0 was observed; a trend that has been observed previously in apoptotic colon fibroblast cells [31], indicating that Econea^®^ might activate apoptosis in human cells. Polyunsaturated TAGs can also accumulate during apoptosis and senescence and can protect cells from oxidative stress that induced membrane damage [37], consistent with the modest TAG accumulation we observed. CERs have been associated with autophagy induction and act as pro-apoptotic messengers by increasing the permeability of the mitochondrial outer membrane, thereby mediating caspases activation [37,38]. A previous proteomics study conducted on gills of the target species *M. galloprovincialis* after two days of Econea^®^ exposure (1 µg/L; 0.003 µM) also showed significant protein alterations in ABC transporters (ABCB/P-glycoprotein-like protein), chromatin (histone H2B), oxidative stress (CuZn superoxide dismutase), immune response (myeloid differentiation factor 88a, C1q-domain-containing protein, putative C1q domain containing protein MgC1q8), and protein biosynthesis (a ribosomal protein) [8]. A more recent study showed that Econea^®^ disrupted synthesis and β-oxidation of FA in lipid metabolism, increasing the synthesis of PCs on the non-target organism zebrafish [39]. Interestingly, in this present study, Econea^®^ also caused depletion in more than 20 PCs, followed by CERs (9). On the other hand, Econea^®^ caused modest depletion (<2-fold change) in CERs C18:1, C22:0, C22:1, C24:0, C24:1, C26:0, PCs C42:0-3, C44:1-5, C46:2-6, C48:6, and TAG C60:4, compared to the control cell (Figure 3A).

Unlike the commercial biocide Econea^®^, compound 1 produced more lipid depletion (71), namely, in more than 30 PCs, followed by SM (10), and CERs (9) (Figure 3B). Compound 1 caused modest depletion (<0.5-fold change) in CERs C14:0, C16:1, C18:0, C18:1, C20:0, C22:1, C24:1, DiHCER C14:0, C16:0, C22:1, DiHDeoxyCERs C16:0, C18:0, C20:0, PCs C42:6, C44:4, C44:5, C44:6, SMs C22:0, C24:0, C26:1, and TAGs C56:8, C58:8 (Figure 3B). However, compound 1 caused an increase of a few PCs (7), followed by TAGs (10). Unlike Econea^®^, compound **1** causes modest accumulation (1.5–3-fold change) in membrane lipids C40:0, C42:0, C44:0-1, C48:6, PCs C22:0, C24-6:1, SMs C54:1, and TAG C58:2. The increases in membrane lipids suggest membrane remodeling, which could be linked to the mechanism of their uptake, indicating that compound **1**, despite being non-toxic at the concentrations tested, may cause changes in lipid homeostasis. Furthermore, a substantial increase in lipids with very long polyunsaturated chains can make the cell membrane more sensitive to oxidative stress [29]. Results also showed that DAGs, which are lipid messengers of immune cells, remained unchanged after compound **1** treatment. On the other hand, the absence of CER accumulation is consistent with the sustained viability of compound **1**-treated cells observed in the MTT assays.

Out of the 71 features initially considered, a two-way ANOVA analysis revealed an accumulation of 13 features and a depletion of 1 feature with Econea^®^ treatment, and 8 features that were depleted when treated with compound **1** (*p*-value < 0.05). This analysis is useful in determining whether changes in one lipid feature correlate with changes in other lipid features, regardless of absolute abundances. One depletive fold change (0.72) for the feature was annotated as PC C32:0 (****). The rest of these features were accumulations with fold changes recorded between 1.19 and 2.48. These features were annotated as PC C28:3 (****), C30:1 (****), C30:0 (****), C32:2 (****), C32:1 (****), C34:3 (****), C34:2 (****), C36:4 (****), C36:3 (****), C36:2 (****), C38:5 (****), C38:4 (****), and SM C16:0 (****) (* adjusted *p*-value < 0.05, ** adjusted *p*-value < 0.01, *** adjusted *p*-value < 0.001, **** adjusted *p*-value < 0.0001). Therefore, further metabolomics and enzymatic studies related to oxidative stress need to be conducted on this human cell line to confirm a possible Econea^®^ cell death pathway.

Unlike the commercial biocide Econea^®^, there were fewer features in compound 1-treated cells that had an adjusted *p*-value < 0.05. The analyzed features were annotated as 6 PC and 1 SM. These eight features had modest depletions with fold change values between 0.66 and 0.89 concerning the control. These features were annotated as PC C30:0 (*), C32:0 (****), C34:2 (****), C38:5 (****), C38:4 (****), C38:3 (**), C40:5 (***), and SM C16:0 (*) (* adjusted *p*-value < 0.05, ** adjusted *p*-value < 0.01, *** adjusted *p*-value < 0.001, **** adjusted *p*-value < 0.0001).

Econea^®^ exposure resulted in a significant accumulation of PCs, which may be related to its toxic effect. Comparatively, there were fewer PCs affected with modest depletions (adjusted *p*-value < 0.05) with compound **1** treatments. In addition, compound **1** did not reduce cell viability at the concentrations tested, making this compound a suitable option to be incorporated into marine coatings to prevent the attachment of marine organisms. In the future, more studies should be addressed to understand the compound **1** mechanism of action.

## 3. Materials and Methods

### 3.1. Synthesis

Compound **1** was obtained by click chemistry from a propargylated chalcone and a glycosyl azide according to our previously described synthesis [19]. The purity was evaluated by high-performance liquid chromatography coupled to an ultraviolet detector (HPLC-UV) through a mobile phase containing water/acetonitrile (30:70 *v*/*v*) with a final adjusted pH to 2.5 and a constant flow rate of 1.0 mL/min in isocratic mode. The injection volume was 10 μL, the C18 column (Fortis Technologies, Neston, UK, 5 μm, 250 × 4.6 mm) was maintained at room temperature of 22 ± 1 °C, and the detection wavelength was set at 250 nm. A purity higher than 95.0% was obtained for compound **1** (Appendix A, Appendix A). Stock solutions of Econea^®^ (purity ≥ 95.0%, Janssen PMP, Beerse, Belgium) and compound **1** (20 mM) were prepared by dissolving each compound into dimethyl sulfoxide (DMSO, Sigma-Aldrich, Burlington, MA, USA) and were stored at −20 °C.

### 3.2. Cell Culture

Immortalized hTERT-RPE-1 cells were obtained from the American Type Culture Collection and maintained in proliferating conditions in Dulbecco’s Modified Eagle Medium/Nutrient Mixture F-12 (DMEM/F12, Corning Inc., Corning, NY, USA) media with 10% of fetal bovine serum (FBS, Sigma-Aldrich) and 1% of penicillin and streptomycin (Corning) at 37 °C until 80% confluence before exposure to treatments.

### 3.3. Mitochondrial Viability Assay

To evaluate the IC_50_ of Econea^®^ and compound **1** in hTERT-RPE-1 cells, a cellular viability assay was performed for each compound using an MTT reagent (Alfa Aesar, VWR, Ward Hill, MA, USA). hTERT-RPE-1 cells were seeded at 4000 cells per well in 96-well plates and were allowed to attach for 24 h. Treatment concentrations of 1, 3, 10, 30, and 100 μM were used with a vehicle control of DMSO. The final concentration of DMSO was held constant at 0.5% across all treatments. After 24 h of treatment, the growth media was removed and replaced with a 9% MTT solution in complete growth media. After 3 h, all the MTT solutions were removed and replaced with 100 μL of DMSO to dissolve formazan crystals. The plate was incubated at 37 °C for 10 min. The absorbance of each well was measured using a Bio-Tek Synergy^TM^ HT plate reader at 550 nm. All experiments were carried out in triplicate for each treatment condition. Viability was determined by comparing the average absorbance signal of treatment to the control. A paired Student’s *t*-test was used to determine statistical significance.

### 3.4. Lipid Extraction

Cells were seeded at 5 × 10^5^ cells per 10 cm plate and allowed to attach for 24 h before the addition of 10 μM Econea^®^, compound **1**, or a DMSO vehicle control (*n* = 3). After a 24 h incubation, cellular medium was removed, and cells were scraped and washed with phosphate-buffered saline (PBS) solution. Extractions were performed by first adding 1 mL of cold PBS to thawed cell pellets with a 30 μL aliquot taken for protein normalization; cells were stored in a −80 °C freezer until analysis. Following that, 3 mL of cold chloroform/methanol (2:1, *v*/*v*) mixture was combined with the remaining 970 μL cell lysate in a homogenizer to increase extraction efficiencies of lipids from cultured cells. Cells were homogenized and centrifuged at 500× *g* at 4 °C for 10 min to facilitate phase separation. A chloroform layer of 1.6 mL in each sample was taken, dried down completely, and resuspended in an internal standard spiked chloroform solution at a normalized volume based on protein content. Internal standards for this study included d9-oleic acid (5 μM), glucosylceramide C17:0 (5 μM), 1,2-distearoyl-sn-glycerol-3-phosphocholine (D70 DSPC, 5 μM), CER C17:0 (1 μM), sphingosine C17:0 (5 μM), sphingomyelin C17:0 (SM, 5 μM), TAG C39:0 and C57:0 (1 μM), and diacylglycerols C14:0 (DAG, 1 μM). Appendix A includes mass-to-charge (*m*/*z*) ratios, RT, and abundance values for each standard lipid species, as well as fold change and *p*-value after sample spiking. All lipid standards were acquired from Avanti Polar Lipids (Alabaster, AL, USA).

### 3.5. Sample Normalization

Protein content was determined by a Bradford protein assay [40]. The 30 μL aliquot was taken from each cell suspension mixed with a lysate buffer and incubated on ice for 1 h. The lysate was centrifuged at 16,000× *g* at 4 °C for 15 min. The aliquot of the lysate was mixed with 1 mL of Coomassie blue, and its protein content was measured at 595 nm using a spectrophotometer. The bovine serum albumin (BSA) standard curve was also prepared to quantify protein content in each sample (Appendix A). Treated and control samples were reconstituted with internal standard spiked in chloroform based on their protein content (Appendix A). Special consideration was taken for the Econea^®^-treated samples that had lower protein content. For that reason, control aliquots were further diluted to be normalized to the lower protein concentrations of the Econea^®^ samples.

### 3.6. Sample Normalization of Liquid Chromatography–Tandem Mass Spectrometry Data

Sample separation and analysis were performed using an Agilent LC-Q-TOF/MS instrument coupled to an Agilent 1260 HPLC system, as described previously by [31,32,33]. Ionization was achieved using a duel Jetstream electrospray source with an ionization voltage of 3500 volts. Source conditions included a drying gas temperature of 350 °C with a flow rate of 12 L/min. A scan range of 50–1700 *m/z* was utilized. Analytical separation was performed using two methods: first using a Luna C5 50 × 4.6 mM 5 μM column for the LC/MS analysis conducted using electrospray ionization under positive mode (+ESI), and second using a Gemini C18 50 × 4.6 mM 5 μM column for the LC/MS analysis conducted under negative mode (-ESI). Mobile phase A was 95% water with methanol, and mobile phase B was 65% isopropanol, 35% methanol, and 5% water. For the analysis under +ESI, the mobile phases contained 0.1% formic acid and 0.1% ammonium formate, while for the analysis under -ESI, the mobile phases contained 0.1% ammonium hydroxide. Both negative mode and positive mode runs utilized the same chromatographic gradient: 5 min, 100% A at 0.1 mL/min; 5–60 min, 100% B at 0.5 mL/min; 60–80 min, 100% A at 0.1 mL/min, with a total run time of 70 min. The sample injection volume used was 10 μL. The LC-MS grade solvents and buffers were purchased from EMD Millipore (Saint Louis, MO, USA).

Agilent MassHunter^TM^ Qualitative Analysis software (version B.06.00) was used for analysis in +ESI and -ESI mode by extracting the corresponding mass-to-charge (*m*/*z*) values for each lipid. The peak areas were manually integrated for the three biological replicates in each treatment. To reduce bias in peak selection, the following criteria were applied: species must fall within an RT threshold of +/− 0.1 min; species must not deviate greater than 10 ppm between replicates, and species must not be present in a blank sample >10% of the signal in the sample. Each peak was integrated separately and considerations of consistency in baseline and shoulder inclusion were included. While manual integration introduced increased bias, automated integration had a higher case of false positives (incorrect peak integration, inconsistent integrations, etc.). Fold changes were calculated by dividing the average raw abundance of the treated sample (i.e., Econea^®^, etc.) by the average raw abundance of the respective control. The heat map was prepared in GraphPad Prism (version 9.3.1). Two-way ANOVA analysis was used to further determine significance of lipid fold changes between treatments and control (* adjusted *p*-value < 0.05, ** adjusted *p*-value < 0.01, *** adjusted *p*-value < 0.001, **** adjusted *p*-value < 0.0001). Spearman correlation plots were prepared in R Studio (2021.09.1) to evaluate the correlation between lipids that had significant changes (adjusted *p*-value < 0.05). The most abundant PCs (C32:0, C38:4, and C44:1) and SMs (C18:0, C24:0, and C26:1) lipid species were further confirmed by LC-MS/MS to assign lipid identities based on known ion fragmentations on METLIN databases (Appendix A) [31,32,33]. Only representative species that were detected at higher ion counts were chosen to help verify the identity of these lipid species. Many detected species were too low of a concentration for meaningful MS/MS experiments.

## 4. Conclusions

Lipidomic approaches were used to evaluate the toxicity of the commercial biocide Econea^®^ and a new promising compound, a triazolyl glycosylated chalcone (compound **1**), on retinal human cells. It was found that Econea^®^, contrary to compound **1**, caused cell mortality at the lowest concentration tested, increasing some lipids related to mitochondrial function, energy storage, and membrane support. These results reinforce the urgent need to develop truly eco-friendly AF agents with lower deleterious impacts on non-target species. Our study demonstrated the use of the lipidomics approach as an effective complementary tool to assess the ecotoxicity of industrial chemicals at lower concentrations that are more environmentally relevant, without the use of live animals. In the future, compound **1** can be incorporated into marine coatings to assess its release into the water through leaching assays. To be effective, the biocide can be incorporated in paints for surface applications such that it can be released minimally, but at a rate necessary to prevent the growth of marine organisms on underwater surfaces. For optimum release and longer effectivity, ablative paints such as controlled depletion polymer technology could be used.

## Figures and Tables

**Figure 1 molecules-27-05247-f001:**
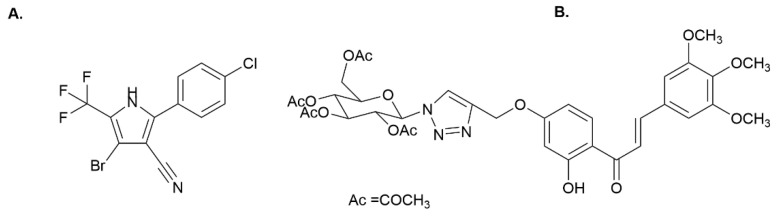
(**A**) Econea^®^ (tralopyril); (**B**) triazolyl glycosylated chalcone (compound **1**).

**Figure 2 molecules-27-05247-f002:**
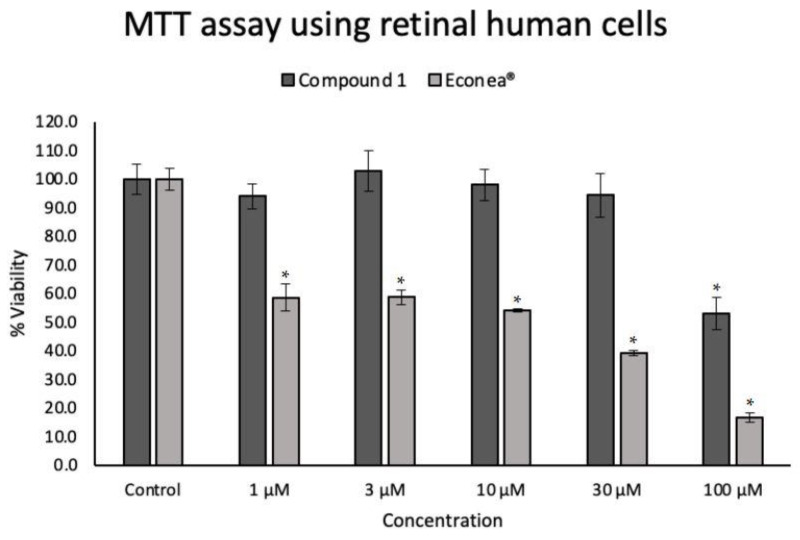
Cytotoxicity evaluation of compound **1**, a promising NIAF, and the emerging biocide Econea^®^. Cell viability was determined by MTT assay in human immortalized retinal pigment epithelial cell line (hTERT-RPE-1) after 24 h of several treatment concentrations. Values are expressed as mean (*n* = 3) ± SD. * *p* < 0.05 compared with the control group.

**Figure 3 molecules-27-05247-f003:**
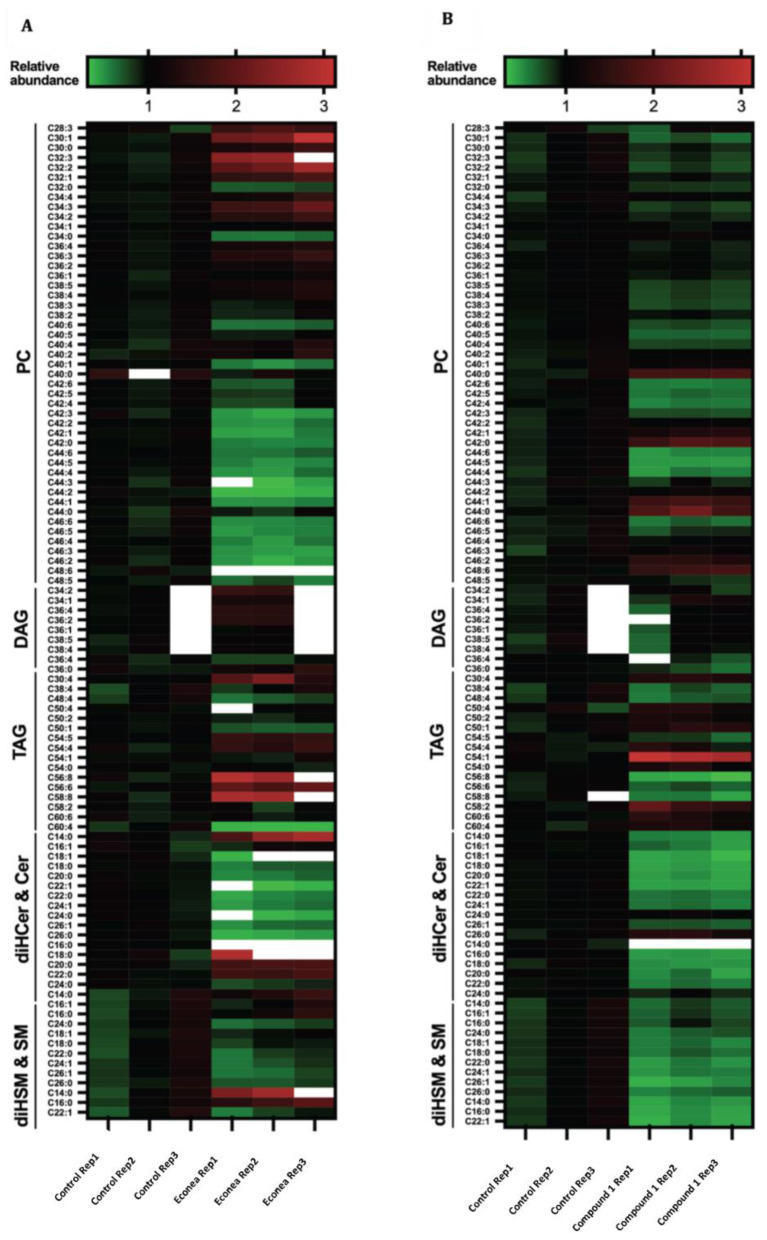
Heat map of differentially expressed lipids in several replicates of control, 10 µM of the Econea^®^ (**A**), and compound **1** (**B**) treated groups for a period of 24 h. White cells represent “not detected” lipids.

## Data Availability

Data will be provided upon request.

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
