# Peer review of "Impact of Tralopyril and Triazolyl Glycosylated Chalcone in Human Retinal Cells’ Lipidome"

_molecules, 2022, doi:10.3390/molecules27165247_

Round 1

Reviewer 1 Report

In this article, titled “Impact of tralopyril and triazolyl glycosylated chalcone in human retinal cells lipidome”, Vilas-Boas et al, aim to characterize the lipid compositions of hTERT-RPE-1 by targeted lipidomics under 10µM of antifouling molecules. This work is useful for the scientific community and the manuscript is well written and presented but needs to answer the following points:

The rationale for the lipidomics analysis of human tissues grown with antifouling biocides is not clear and needs to be explained. Why lipids could be impacted by these molecules? What are the putative interactions of antifoulings with amphiphilic plasma membrane lipids?

Line 85: Need to rephrase as TAG are not involved directly in membrane integrity.

Why other phospholipids are not included in the paper like PE, PG, PI? PI is a key lipid involved in many biological functions (signaling) and should be included.

Figure 3, for compound 1, it needs to be compound 1, rep 1, … and not compound1, 2… as it’s confusing for readers.

A discussion about membrane fluidity under antifouling molecules would be an asset for the paper. Likewise, did the authors have measured global fatty acids composition by GC-FID? As all lipid classes are not measured in this work a global idea of the fatty acid composition should give more information on the studied biocidals effect on human cells.

Author Response

1-Comments and Suggestions for Authors

In this article, titled “Impact of tralopyril and triazolyl glycosylated chalcone in human retinal cells lipidome”, Vilas-Boas et al, aim to characterize the lipid compositions of hTERT-RPE-1 by targeted lipidomics under 10µM of antifouling molecules. This work is useful for the scientific community and the manuscript is well written and presented but needs to answer the following points:

  • The rationale for the lipidomics analysis of human tissues grown with antifouling biocides is not clear and needs to be explained. Why could lipids be impacted by these molecules? What are the putative interactions of antifoulings with amphiphilic plasma membrane lipids?

RESPONSE:

First, we would like to thank the reviewers for their enthusiasm about our paper and useful suggestions that improved our manuscript.

In the revised paper, we clarified the rationale behind using lipidomics on page 3,  justifying the lipidomics analysis of human tissues grown in media with antifouling biocides: “Lipids are relevant cellular components for assessing biological effects of compounds that have high n-octanol-water partition coefficient (log Kow), such as the AF biocides, because lipids play a role in the bioaccumulation of lipophilic compounds.  In addition, AF biocides can disrupt membranes, lipid metabolism and alter the fluidity and permeability of cell membranes of target and non-target organisms [17,20]. Biocides can impact human health not only through the food chain but also by direct exposure during boating maintenance, causing skin allergies, eye irritations, and respiratory disorders.”

  • Line 85: Need to rephrase as TAG are not involved directly in membrane integrity.

RESPONSE:

The sentence was corrected as suggested: “For example, ceramides (CERs) are usually linked to apoptosis while other lipids, such as phosphatidylcholines (PCs) and triacylglycerols (TAGs) are involved in membrane integrity and structural support and storage of excess fatty acids, respectively”.

  • Why other phospholipids are not included in the paper like PE, PG, PI? PI is a key lipid involved in many biological functions (signaling) and should be included.

RESPONSE:

For our analysis we have chosen to monitor lipid species that we believe would be significantly impacted at subtoxic treatment conditions. For this reason, we targeted species such as ceramides and TAGs as these are common species that respond during cellular stress. We also choose to analyze PCs as these species are a major component of the cellular membrane. And has been updated on page 5 of the main text.

  • Figure 3, for compound 1, it needs to be compound 1, rep 1, … and not compound1, 2… as it’s confusing for readers.

RESPONSE:

We thank the reviewer for noticing this; the subtitles have been corrected.

  • A discussion about membrane fluidity under antifouling molecules would be an asset for the paper. Likewise, did the authors have measured global fatty acids composition by GC-FID? As all lipid classes are not measured in this work a global idea of the fatty acid composition should give more information on the studied biocidals effect on human cells.

RESPONSE:

We included a discussion about the lipophilic nature of biocides and their interactions with the lipid bilayer of cellular membranes on page 3. We have also expanded the discussions on the fatty acids measured in this on page 5. The effect of these biocides on membrane fluidity is an interesting research question; however, our current data are limited to be able to make conclusion about membrane fluidity without much speculations. We will consider this aspect in our future experimental designs.

Reviewer 2 Report

The authors submitted a manuscript describing human retinal cell characterization following treatment with a commercial biocide and an experimental biocide.  This work evaluated the toxicity of the biocides on the selected cell line. The need for environmentally friendly solutions is great in the marine industry, and the work to evaluate biocide impact on the environment will aid future biocide development. There are several areas where the manuscript must be improved to increase clarity of the reported results.  The manuscript lacked a comparison of biocide effectiveness, presumably less concentration is needed with more effective solutions.  There was no inclusion of the potential of the biocides to be covalently bound to a coating verse slow release from a coating.  The clearest results are from the MTT assay, which shows clear sensitivity to Econea at the lowest dose examined.  The authors lacked explanation of clear goals for the lipidomic analysis. They offered comparison of systemic changes in marine organisms as a driving factor to explore the lipidome of a single cell line.  There was no data presented to indicated where the population of the cells were in their cell cycles (no synchronization was attempted).  Lipid profiles will change based on cell cycle.  The cell lipidomic data interpretation often differs from that from lipid profiles of organism biofluids or tissues.  The experiments described in the manuscript are non-targeted lipidomic assays, not targeted analysis.  The experiments collect a range of features for set windows and not targeted collection of specific species.  The author use of up-regulated and down-regulated species is not appropriate for metabolites/lipids.  The regulation is better used in genomics, and metabolomics involves with compound abundances (relative or absolute).  The authors included heavy labeled standards in the experiment; however, it was not clear how the standards were used in the study (QC purposes or conversion to absolute concentration).  With the limited number of samples (n=3) statistics can be challenging, inclusion of correlation analysis would help assess the group variance. The following are additional suggestion to improve the manuscript:

- line 38, underwater building structures likely don’t have an increased fuel consumption issue like the water vehicles.

- the manuscript would be improved with addition of short description of fouling progression in the marine environment and where in the progression the biocides are effective.

- Figure 2B, needs labels and scale bar.

- The authors should use “identified” metabolites for when feature is matched to an authentic standard, else use “annotated”.

- Lines 142-143, mentioned FAs eluting first, but no FAs were included in the discussion nor the supplemental table.

- Lines 150-151, the “<10% abundance in the blank sample” is a feature filter and did not impact feature annotation.

- If used only used accurate mass in annotation, then PE and PC lipids would be difficult to distinguish.

- Lines 151-155, this statement is unclear about quantitative analysis and percent of lipids detected and annotated. Quantitative analysis would suggest the relative abundance were converted to absolute abundances with use of standards.

- Line 156, not clear how fold change was calculated. Author should use log2 average FC.  Not clear if 71 selected lipids were compound that changed in response to Econea, compound 1, either, or both. 

Figure 3, clarify what the white blocks represent. Also combine the sub-figures into one as the controls are the same. Compound 1, 2, 3 are confusing since comparison established between Econea and Compound 1.

Lines 165-166, “causes” is too strong to use here better to use associated. These lines refer to numbers of lipids, while figure 4 presents percentages. Figure 4 is not needed in the main text, as the main points are observable in the heatmap in figure 3.

Lines 170-174, unclear if there is significance to grouping of different lipids in higher abundance.

Typical naming convention for the lipids TAG(30:4), PC(36:3), etc. The lists of compounds in the text need to be clearer and list each compound individually. “/” are used to indicate knowledge of acyl chains which is not the context in the manuscript.

Line 330, were there steps to remove bias from manual integration of the LC-MS peaks?

Line 309, the text mentioned MSMS was collected although no data was included.  Supplemental would be an excellent place for this data. With MSMS information some chain information should be possible.  Text above previously stated the annotations were based on accurate mass.

Table S1, “Full Change” change to “Fold Change”. C34:0 PC and others PPM errors are not <5. Not appropriate to list 0 abundance in blank, should establish limit of detection (LOD) and state below LOD. Need to define all abbreviations.

The TAG should also show up as ammonia adducts, and possibly you would have more detected TAG if included those.

Line 313, “comprehensibly evaluate” is too strong of a statement based on included data.

Author Response

2-Comments and Suggestions for Authors

The authors submitted a manuscript describing human retinal cell characterization following treatment with a commercial biocide and an experimental biocide. This work evaluated the toxicity of the biocides on the selected cell line. The need for environmentally friendly solutions is great in the marine industry, and the work to evaluate biocide impact on the environment will aid future biocide development. There are several areas where the manuscript must be improved to increase clarity of the reported results.  The manuscript lacked a comparison of biocide effectiveness, presumably less concentration is needed with more effective solutions. 

RESPONSE:

We thank the reviewer for suggesting this improvement to the rational of the paper. We introduced the antifouling effectiveness of Econea (EC50 = 4.012 μM; 1.40 µg/mL) against the mussel Mytilus galloprovincialis and already described in previous works in order to strengthen the importance of compound 1(EC50 = 3.28 μM; 2.43 µg/mL) as a new promising antifouling agent. This information can be found on page 2 of the main text.

  • There was no inclusion of the potential of the biocides to be covalently bound to a coating versus slow release from a coating.  

RESPONSE:

The following line was included in the conclusion section, on page 12: “In the future, compound 1 can be incorporated into marine coatings to assess its release to water through leaching assays and the effectiveness of the generated coating evaluated.”

  • The clearest results are from the MTT assay, which shows clear sensitivity to Econea at the lowest dose examined.  The authors lacked explanation of clear goals for the lipidomic analysis. They offered comparison of systemic changes in marine organisms as a driving factor to explore the lipidome of a single cell line. 

RESPONSE:

We thank the reviewer for suggesting this improvement to the rationale of the paper. We introduced a sentence on page 3 justifying the lipidomics analysis of human tissues grown with antifouling biocides: “Lipids are relevant cellular components for assessing biological effects of compounds that have high n-octanol-water partition coefficient (log Kow), such as the AF biocides, because lipids play a role in the bioaccumulation of lipophilic compounds.  In addition, AF biocides can disrupt membranes, lipid metabolism and alter the fluidity and permeability of cell membranes of target and non-target organisms [17,20]. Biocides can impact human health not only through the food chain but also by direct exposure during boating maintenance, causing skin allergies, eye irritations, and respiratory disorders.”

 There was no data presented to indicated where the population of the cells were in their cell cycles (no synchronization was attempted).  Lipid profiles will change based on cell cycle.  

RESPONSE:

The reviewer is correct that the lipid compositions of cells change as they progress in their cell cycle (Atilla-Gokcumen et al. Cell, 2014). In this study, we did not synchronize cells at a particular cell cycle stage as a diverse population is a better representative of what happens at the organismal level when target and non-target organism are exposed to these compounds. However, exposure of synchronized cells would be an interesting direction to elucidate the detailed mechanisms of toxicity of these compounds. We added this limitation of the study in the discussion.

  • The cell lipidomic data interpretation often differs from that from lipid profiles of organism biofluids or tissues.  The experiments described in the manuscript are non-targeted lipidomic assays, not targeted analysis.  The experiments collect a range of features for set windows and not targeted collection of specific species.  The author use of up-regulated and down-regulated species is not appropriate for metabolites/lipids.  The regulation is better used in genomics, and metabolomics involves compound abundances (relative or absolute).  

RESPONSE:

We agree the terms “up and down regulated” were used inappropriately and we have corrected the text in our manuscript to refer to these results as “fold-changes”.

  • The authors included heavy labeled standards in the experiment; however, it was not clear how the standards were used in the study (QC purposes or conversion to absolute concentration).  With the limited number of samples (n=3) statistics can be challenging, inclusion of correlation analysis would help assess the group variance.

RESPONSE:

We have improved the main text to better explain the role of the internal standards in our experimental analysis. This can be found on page 5 of the main text. We have also updated our analysis to include a more powerful two-way ANOVA analysis and a Spearman correlation to a stronger statistical analysis of features. This can be found on pages 7-9 of the main text.

  • line 38, underwater building structures likely don’t have an increased fuel consumption issue like the water vehicles.

RESPONSE:

We thank the reviewer for this correction. The sentence was modified to: “Fouling of underwater surfaces of ships and vessels has many economic impacts for the marine industry, which has become the primary cost associated with the increase of fuel consumption attributed to increased frictional drag [1]”.

  • the manuscript would be improved with addition of short description of fouling progression in the marine environment and where in the progression the biocides are effective.

RESPONSE:

The following paragraph was included in the introduction section, on page 2: “Marine biofouling corresponds to a successive accumulation of a wide range of colonizer species (e.g. bacteria, diatoms, macroalgae, tunicates, barnacles, mussels, and tubeworms) into surfaces immersed in seawater. To prevent or inhibit the settlement and growth of marine organisms on underwater surfaces AF biocides are coated on the surface and must be continually released at the surface -water interface at a rate necessary to generate a toxic concentration within the surface boundary layer.”

  • Figure 2B, needs labels and scale bar.

RESPONSE:

We agree with the reviewer; therefore, figure 2B was removed from the article.

  • The authors should use “identified” metabolites for when feature is matched to an authentic standard, else use “annotated”.

RESPONSE:

We agree, thanks for the comment. The statement was modified accordingly in the revised manuscript. Also, the most abundant species were identified by MS/MS and included in the supplementary (Table S3).

  • Lines 142-143, mentioned FAs eluting first, but no FAs were included in the discussion nor the supplemental table.

RESPONSE:

Fatty acids were included in our analysis, however we did not observe significant changes of these species. For this reason, we did not include a discussion in our original manuscript. However, we agree with the reviewer that this is important. We have added a brief discussion on page 5.

  • Lines 150-151, the “<10% abundance in the blank sample” is a feature filter and did not impact feature annotation.

RESPONSE:

We agree with Reviewer #2 in regard to this comment. We had a selection criterion that said potential features must have an abundance higher than 10% of “blank” signal. This however is not related to our annotation criteria. We have updated the main text on page 5.

  • If used only used accurate mass in annotation, then PE and PC lipids would be difficult to distinguish.

RESPONSE:

We thank the reviewers for this point. We neglected to explain in the text that the LC-MS method we used in this study was previously developed and validated. To validate our detection of PCs in this study and to differentiate them from PEs, we conducted MS/MS experiments to obtain fragmentation for representative m/z’s and confirmed their annotation. The fragments are now included in the SI Table S3. These fragments are shared between all three features and are consistent with the phosphocholine headgroup. We include additional discussion of our MS/MS experiment on pages 11 and 12.

  • Lines 151-155, this statement is unclear about quantitative analysis and percent of lipids detected and annotated. Quantitative analysis would suggest the relative abundance were converted to absolute abundances with use of standards.

RESPONSE:

Our analyses in this study are comparative, we thank the reviewer for catching this error. We have revised our descriptions accordingly.

  • Line 156, not clear how fold change was calculated. Author should use log2 average FC.  Not clear if 71 selected lipids were compound that changed in response to Econea, compound 1, either, or both.

RESPONSE:

We thank the reviewer for this suggestion. However, we prefer to report the fold changes in their non-transformed form. We believe the relative comparison of the changes in lipids will be easier in this form.

The 71 lipids changed in response to both compounds, we have clarified this in the text.

  • Figure 3, clarify what the white blocks represent. Also combine the sub-figures into one as the controls are the same. Compound 1, 2, 3 are confusing since comparison established between Econea and Compound 1.

RESPONSE:

We thank the reviewer for noticing the figure lapse. The subtitles have been corrected. The white blocks represent lipids that were non detected and were also clarified on the subtitle.

  • Lines 165-166, “causes” is too strong to use here better to use associated. These lines refer to numbers of lipids, while figure 4 presents percentages. Figure 4 is not needed in the main text, as the main points are observable in the heatmap in figure 3.

RESPONSE:

We thank the reviewer for their suggestion. The statement was modified accordingly in the revised manuscript.

 Figures 4 and 5 were removed and the wording expression corrected.

  • Lines 170-174, unclear if there is significance to grouping of different lipids in higher abundance.

RESPONSE:

We systematically targeted species with C14-C58 chain length and saturations, and 1-3 unsaturation’s.

  • Typical naming convention for the lipids TAG (30:4), PC (36:3), etc. The lists of compounds in the text need to be clearer and list each compound individually. “/” are used to indicate knowledge of acyl chains which is not the context in the manuscript.

RESPONSE:

We appreciate this comment and we have updated the main text to follow this nomenclature guideline.

  • Line 330, were there steps to remove bias from manual integration of the LC-MS peaks?

RESPONSE:

To reduce bias in our peak selection we followed the following criteria. Species must not be present in a blank sample >10% of the signal in the sample, species must fall within a retention time threshold +/- 0.1 minutes, species must not deviate greater than 5 ppm between replicates. We introduced bias by integrating each peak separately, however we made efforts to reduce this by being stringent on how each peak was integrated including considerations of consistency in baseline and shoulder inclusion. While manual integration could increase bias, automated integration had a higher case of false positives (incorrect peak integration, inconsistent integrations, etc.). This discussion has been included on page 11 of the main text.

  • Line 309, the text mentioned MSMS was collected although no data was included.  Supplemental would be an excellent place for this data. With MSMS information some chain information should be possible.  Text above previously stated the annotations were based on accurate mass.

RESPONSE:

The MS/MS data were included in the supplementary information (Table S1).

  • Table S1, “Full Change” change to “Fold Change”. C34:0 PC and others PPM errors are not <5. Not appropriate to list 0 abundance in blank, should establish limit of detection (LOD) and state below LOD. Need to define all abbreviations.

RESPONSE:

Thanks for the comment. Table S1 was corrected.

  • The TAG should also show up as ammonia adducts, and possibly you would have more detected TAG if included those.

RESPONSE:

We have indeed looked for the ammonium adduct of TAGs. We have updated this on page 5 of the main text.

  • Line 313, “comprehensively evaluate” is too strong of a statement based on included data.

RESPONSE:

We agree, thanks for the comment. The statement was modified accordingly in the revised manuscript.

Reviewer 3 Report

The manuscript adopts a targeted lipidomics approach to systematically evaluate the developmental toxicity mechanism of tralopyril and triazolyl glycosylated chalcone in human retinal cells, which can provide a reference for green fungicides. The research is innovative and practical. The obtained data can support the results. But before it can be accepted by Molecules, it needs some revisions.

1 Error bars and significance analyses are missing from Figure 4 and Figure 5.

2 Reference format needs to be carefully revised.

3 How does triazolyl glycosylated chalcone availability compare to tralopyril, and if used as an antibacterial additive in coatings, what characteristics are required to minimize losses in water?

Author Response

3- Comments and Suggestions for Authors

The manuscript adopts a targeted lipidomics approach to systematically evaluate the developmental toxicity mechanism of tralopyril and triazolyl glycosylated chalcone in human retinal cells, which can provide a reference for green fungicides. The research is innovative and practical. The obtained data can support the results. But before it can be accepted by Molecules, it needs some revisions.

RESPONSE:

We thank the reviewer for their enthusiasm about our paper.

  • Error bars and significance analyses are missing from Figure 4 and Figure 5.

RESPONSE:

Figure 4 and 5 represent the number of species hence there are no error bars associated. However, for not being pertinent to the understanding of the results, figure 2B was removed from the article.

  • Reference format needs to be carefully revised.

RESPONSE:

We have carefully went over the references and revised where needed.

  • How does triazolyl glycosylated chalcone availability compare to tralopyril, and if used as an antibacterial additive in coatings, what characteristics are required to minimize losses in water?

RESPONSE:

The following line was included in the conclusion section: “In the future, compound 1 can be incorporated into marine coatings to assess its release to water through leaching assays and the effectiveness of the generated coating. To be effective, the biocide can be incorporated in paints for surface application such that it can be released minimally, but at a rate necessary to prevent growth of marine organisms on the underwater surfaces. For optimum release and longer effectivity, ablative paints such as controlled depletion polymer technology could be used.”

Round 2

Reviewer 1 Report

The authors have answered all my questions. 

Author Response

1-Comments and Suggestions for Authors

The authors have answered all my questions. 

RESPONSE:

We thank the reviewer for the comments and suggestions that improved our manuscript.

Reviewer 2 Report

The manuscript has been improved by the authors; however, several issues still need to be addressed. 

The annotations need to be reviewed by an expert.  As an example, the feature 643.5302_61.26 was annotated as an M+H ion of DAG(36:0), however according to LipidsMaps database the DAG(36:0) [M+H]+ would be 625.5765 m/z and the [M+NH4]+ is 642.6031 m/z neither of which match the m/z to support the annotation by the authors.

The inclusion of the MSMS data was appreciated, however this only supported the annotation of six compounds. The rest of the annotations are based on accurate mass matches, the manuscript would be greatly enhanced by additional MSMS confirmed annotations. It is advisable to include the entire dataset in the supplemental so in the future other researchers can analyze the dataset in different ways.  Table S2 looks like mostly positive mode data, the authors should clarify.  Negative mode collection was included in the methods however no data was shown or discussed.  There would be no need to include the methods if the data is not part of the scientific story.

In the conclusion, “Targeted lipidomic approaches” were not described in the manuscript.  Furthermore, the study design lacks the experiments to properly evaluate how the lipidomic signals are related to level of toxicity of the antifouling agent.  The study would have been greatly improved by evaluating the 100uM concentration of both antifouling agents, where both compounds had toxic effects according to the MTT assay. There was no evidence that the effects of Econea and compound 1 on the cell line is result the same cellular mechanism; therefore, it is difficult to compare the experiments (shown in figure 3).   

Most of the lipids in table S2 eluted between 55-71 min with overlap of the lipid classes; thus, it is not clear how in line 183 the standards were used to verified RT of each lipid class. Additionally, not clear how the standards aided in annotation since none of the standards matched compound in the dataset listed in table S2. Line 180-182, unclear how the standards added at reconstitution step would serve as a QC for lipid extraction. The normalized peak areas for the standards should be included so the reader can assess the quality of the data processing and normalization.  It would be good to include the Bradford results for readers to understand the normalization procedure, since the samples were adjusted by dilution prior to analysis, presumably during reconstitution. It is not clear is the source of the cell suspension for the Bradford assay. Was it the cell pellet from the lipid extraction? Line 356 refers only to a cell suspension which is unclear.

The correlation matrix is frequently more useful to assess sample to sample variation.  It is unclear if figures 4 and 5 are necessary, as main message is that the features annotated as PC were similar to each other with one exception, and this result would not be significant.

In figure 3, unclear why there are differences in the control data between the two sub-figures. Was the data treated different for each set of data?

Author Response

2-Comments and Suggestions for Authors

The manuscript has been improved by the authors; however, several issues still need to be addressed. 

  • The annotations need to be reviewed by an expert.  As an example, the feature 643.5302_61.26 was annotated as an M+H ion of DAG(36:0), however according to LipidsMaps database the DAG(36:0) [M+H]+would be 625.5765 m/z and the [M+NH4]+ is 642.6031 m/z neither of which match the m/z to support the annotation by the authors.

RESPONSE: We apologize for the confusion here. Based on previous protocols established in this lab we used [M-OH]+ adducts for many of the DAGs that we analyzed, with only 2 being [M+H]+. Table S4 in the supplemental information and page 5 (lines 190-194) of the manuscript have both been updated to reflect these changes.

  • The inclusion of the MSMS data was appreciated, however this only supported the annotation of six compounds. The rest of the annotations are based on accurate mass matches, the manuscript would be greatly enhanced by additional MSMS confirmed annotations. It is advisable to include the entire dataset in the supplemental so in the future other researchers can analyze the dataset in different ways. 

RESPONSE:

Following the previous revision in response to reviewer 2, a Two-way ANOVA analysis was used to further determine significance of lipid fold changes between treatments and control, which allowed us to observe significant changes in only two lipid classes, PCs and SMs. For that reason, the three most abundant PCs (C32:0, C38:4, and C44:1) and SMs (C18:0, C24:0, and C26:1) lipid species were further confirmed by LC-MS/MS allowing us to confirm lipid identities of these two classes (Table S5). In the future, we will consider analyzing all lipid classes to obtain more information. We are limited to this form of analysis, however, since low abundance species would not give meaningful fragmentation data. We thank the reviewer for this advice. Improved discussion on this can be found on page 10 (lines 384-390)

  • Table S2 looks like mostly positive mode data, the authors should clarify. Negative mode collection was included in the methods however no data was shown or discussed.  There would be no need to include the methods if the data is not part of the scientific story.

RESPONSE:

All ceramide classes were analyzed in negative mode. To clarify, a column with each adduct used was added to Table S1. We have also corrected the following sentence in the manuscript: PCs, SMs, dHSMs, were detected in the protonated form [M+H]+. Some DAGs were detected as [M+H]+ however most were detected as [M-OH]+. All TAGs were detected with the ammonium adduct [M+NH4]+. The rest of the lipids CERs, DiHCER, DeoxyCER, and DiHDeoxyCER were identified as [M-H]-. This change can be seen on page 5 (lines 190-194) of the main text.

  • In the conclusion, “Targeted lipidomic approaches” were not described in the manuscript.  

RESPONSE:

In this lipidomic study, we look for a targeted collection of specific species. Furthermore, several known lipid species were added to our samples to serve as internal standards and to ensure a stronger identification of the various lipids present in the samples. However, the expression has been rectified for “Lipidomic approaches” on page 10 (line 393) of the main text.

  • Furthermore, the study design lacks the experiments to properly evaluate how the lipidomic signals are related to level of toxicity of the antifouling agent. The study would have been greatly improved by evaluating the 100uM concentration of both antifouling agents, where both compounds had toxic effects according to the MTT assay. There was no evidence that the effects of Econea and compound 1 on the cell line is result the same cellular mechanism; therefore, it is difficult to compare the experiments (shown in figure 3).   

RESPONSE:

We agree that a higher concentration would trigger a more intense lipid alteration. However, we wanted to observe lipid regulation of a more realistic stimulus, with a concentration closer to the effective concentration of both compounds against M. galloprovincialis (Compound 1 EC50 = 3.28 μM; (Econea EC50 = 4.012 μM;).

  • Most of the lipids in table S2 eluted between 55-71 min with overlap of the lipid classes; thus, it is not clear how in line 183 the standards were used to verified RT of each lipid class.

RESPONSE: To clarify the role of internal standards in this study, we revised the description in the manuscript. Internal standards (IS) were used in the lipidomics analysis as quality controls; we used IS to monitor for retention time drift over the course of multiple analyses during the day, any drop in sensitivity, and changes in accurate mass measurements. This has been updated on page 5 (lines 179-185) of the main text.

  • Additionally, not clear how the standards aided in annotation since none of the standards matched compound in the dataset listed in table S2.

RESPONSE:

The internal standards used were not used for annotation but for quality controls (see above comment in response to #6). Annotation was done for most species by exact mass (<10 ppm error) and for a select number of compounds we used MS/MS based fragmentation for further confirmation. MS/MS experiments cannot be done on species which had low abundances in our samples because they would not produce meaningful fragmentation. Fold change and p-value of the standards added to the samples before lipid annotations were included to table S1. Only a single IS was used for each lipid class included in our analysis. We have included discussion to help clarify this on pages 5 (lines 179-184) and 10 (lines 384-390) of the main text. 

  • Line 180-182, unclear how the standards added at reconstitution step would serve as a QC for lipid extraction. The normalized peak areas for the standards should be included so the reader can assess the quality of the data processing and normalization. 

RESPONSE:

We thank the reviewer for noticing this; the sentence has been corrected. Internal standards were not used as a quality control for lipid extraction, but only as a quality control for our LC-MS analysis. We also did not use our internal standards to normalize our data but choose to use raw abundances of each species for any fold change and statistical analysis. This can be seen on page 10 (line 381) of the main text. We have also added our entire dataset as a table (Table S3) in the SI.

  • It would be good to include the Bradford results for readers to understand the normalization procedure, since the samples were adjusted by dilution prior to analysis, presumably during reconstitution.

RESPONSE:

We thank the reviewer for suggesting this improvement to the rational of the paper. A calibration curve (Figure S2) and the remaining normalization values (Table S3) were added to the supplementary information.

  • It is not clear is the source of the cell suspension for the Bradford assay. Was it the cell pellet from the lipid extraction? Line 356 refers only to a cell suspension which is unclear.

RESPONSE: When cells were collected as a suspension in 1 mL of PBS we take a 30 μL aliquout for protein normalization. We have updated the main text on page 10 under the protein normalization section and added the remaining information as a table (Table S2) to help clarify this. We have improved this discussion on page 9 (lines 339-344) of the main text.

  • The correlation matrix is frequently more useful to assess sample to sample variation.  It is unclear if figures 4 and 5 are necessary, as main message is that the features annotated as PC were similar to each other with one exception, and this result would not be significant.

RESPONSE:

Figures 4 and 5 were removed from the manuscript.

  • In figure 3, unclear why there are differences in the control data between the two sub-figures. Was the data treated different for each set of data?

RESPONSE:

Two 96-well plates were used to assess cell viability for compound 1 and Econea, therefore 2 controls were also used. The data were not treated differently.

Round 3

Reviewer 2 Report

The authors satisfactorily addressed my concerns.  The following are a few minor points to refine the manuscript further.

Figure 2 was cutoff in the pdf I received.

Line 180, “40 to” change to 40- to.

The [M-OH]+ is usually referred to as [M-H2O+H]+. Should be revise in Line 182 and the supplementary table.

Lines 250-253 and 260-261, not clear what the (****) represent.  Need to define on first use.

Line 256-257, needs revision to clarify statement.

m/z should be italicized.

Author Response

2-Comments and Suggestions for Authors

The authors satisfactorily addressed my concerns.  The following are a few minor points to refine the manuscript further.

  1. Figure 2 was cutoff in the pdf I received.

RESPONSE: Figure 2 was rectified.

  1. Line 180, “40 to” change to 40- to.

RESPONSE: The expression was rectified.

  1. The [M-OH]+ is usually referred to as [M-H2O+H]+. Should be revise in Line 182 and the supplementary table.

RESPONSE: We appreciate the suggestions but would like to maintain the same terminology according to our previous publications.

  1. Lines 250-253 and 260-261, not clear what the (****) represent.  Need to define on first use.

RESPONSE: The statistical terminology was explained on the point 3.6. The following quote “(* Adjusted p-value <0.05, ** Adjusted p-value <0.01, *** Adjusted p-value <0.001, **** Adjusted p-Value <0.0001)” was included after the in lines 250-253 and 260-261

  1. Line 256-257, needs revision to clarify statement.

RESPONSE: The quote was clarified.

  1. m/z should be italicized.

RESPONSE: The term was rectified in the main manuscript and supplementary.